# Human Periodontal Ligament Stem Cells (hPDLSCs) Spontaneously Differentiate into Myofibroblasts to Repair Diabetic Wounds

**DOI:** 10.3390/bioengineering11060602

**Published:** 2024-06-12

**Authors:** Yuxiao Li, Qi Su, Zhaoyu Tao, Xiang Cai, Yueping Zhao, Zhiying Zhou, Yadong Huang, Qi Xiang

**Affiliations:** 1Institute of Biomedicine and Guangdong Provincial Key Laboratory of Bioengineering Medicine, Jinan University, Guangzhou 510632, China; lyx_edison@163.com (Y.L.); su18650872988@163.com (Q.S.); taozhaoyu2014@163.com (Z.T.); 16626412540@163.com (X.C.); tydhuang@jnu.edu.cn (Y.H.); 2School of Stomatology, Jinan University, Guangzhou 510632, China; 13450488226@163.com (Y.Z.); dentistchow@163.com (Z.Z.); 3The First Affiliated Hospital of Jinan University, Guangzhou 510632, China

**Keywords:** human periodontal ligament stem cells (hPDLSCs), myofibroblastic differentiation, diabetic wounds

## Abstract

Advanced glycation end product (AGE) accumulation due to diabetes causes vascular and neurological lesions, delaying healing. The use of stem cells could overcome these problems. Although many studies have shown the potential beneficial effects of stem cell therapies in the treatment of chronic and refractory skin ulcers, their delivery methods are still under investigation. Human periodontal ligament stem cells (hPDLSCs) can spontaneously differentiate into myofibroblasts in specific cultures; therefore, they have the potential to effectively treat diabetic wounds and may also have applications in the field of medical cosmetics. The myofibroblastic differentiation ability of hPDLSCs in the presence of AGEs was evaluated by the expression of α-SMA and COL1A1 using RT-qPCR and WB technology. Wound healing in diabetic mice, induced by streptozotocin (STZ) and assessed using H&E staining, Masson staining, and immunohistochemical (IHC) and immunofluorescence (IF) staining, was used to validate the effects of hPDLSCs. In the wound tissues, the expression of α-SMA, COL1A1, CD31, CD206, iNOS, and vimentin was detected. The findings indicated that in H-DMEM, the expression of COL1A1 exhibited a significant decrease, while α-SMA demonstrated an increase in P7 cells, ignoring the damage from AGEs (*p* < 0.05). In an STZ-induced diabetic C57BL/6J mice whole-skin defect model, the healing rate of the hPDLSCs treatment group was significantly higher than that in the models (on the 7th day, the rate was 65.247% vs. 48.938%, *p* < 0.05). hPDLSCs have been shown to spontaneously differentiate into myofibroblasts in H-DMEM and resist damage from AGEs in both in vivo and in vitro models, suggesting their potential in the field of cosmetic dermatology.

## 1. Introduction

The effective treatment of diabetic wound healing is a global challenge. Hyperglycemia can induce the hyperglycation of proteins and lead to the generation of advanced glycation end products (AGEs) [1], eventually causing blood vessels to break down. Targeting the structural elements of the connective tissue matrix or basement membrane, AGEs accelerate the development of diabetes mellitus and its consequences and alter the structural integrity of blood vessel walls and the basement membrane through receptor-independent pathways, which can lead to wound or gangrene development [2,3]. In addition, chronic hyperglycemia affects the wound-healing process, resulting in delayed skin wound healing [4]. Delayed or even non-healing of wounds in diabetic patients causes many difficulties in clinical treatment. At present, the treatment of diabetic ulcers mainly focuses on local debridement and interventional therapy. Increasing numbers of studies are being devoted to finding suitable methods to improve this phenomenon, and stem cell therapy, as an advanced treatment method, and in light of the high shrinkage activity and collagen secretion and high expression of alpha smooth muscle actin (α-SMA) myofibroblast-like cells, has attracted much attention.

Myofibroblasts (MFs), as a key factor in wound healing and tissue repair, secrete collagen, fibronectin, and other extracellular matrix components to seal wounds and also express α-SMA to promote wound healing [5]. MFs represent transient phenotypes closely associated with connective tissue healing [6], while α-SMA can be used as an important recognition component of these cells without single markers [7,8]. MFs can be produced from a variety of cellular sources, including fibroblasts, pericytes, SMC, and mesenchymal stromal cells (MSCs) [9]. MFs derived from other cells have been shown to produce more contractile activity than proto-myofibroblasts in the absence of changes in the expression of other contractile proteins [10]. Therefore, a number of studies have attempted to use MSCs and other cells as therapeutic options for diabetic wound repair. Most mesenchymal stem cells used in previous studies for therapeutic purposes were derived from bone marrow and adipose tissue [11,12]. However, the difficulty in sampling and the degree of damage to the host greatly limit their clinical application. A better source of suitable cells is therefore needed.

Human periodontal ligament stem cells (hPDLSCs) are undifferentiated mesenchymal cells found in the periodontal ligament that can differentiate into osteoblasts, adipocytes, chondrocytes, and other cell types in vitro. The ability to produce cementoid tissue and fibers is retained during in vitro transplantation [13]. hPDLSCs have higher growth potential and stronger proliferation ability than human bone marrow-derived MSCs and human dental pulp stem cells (hDPSCs) [14]. Under appropriate stimulation, hPDLSCs can express fibroblast-like genes (e.g., Collagen-1 (COL1), COL3, fibroblast-specific protein 1) and elastin, indicating their fibroblastic activity [15]. Compared with human bone marrow mesenchymal stem cells, hPDLSCs can form more ordered collagen fiber bundle structures and produce more extracellular matrix and collagen [16].

Equally important is the fact that hPDLSCs are prone to non-invasive separation from periodontal tissue during standard tooth scaling and root planing procedures [17]. They can be collected as waste biological samples in dental clinics [18], and the process of collecting hPDLSCs from the oral cavity is minimally invasive. The wound heals without scarring, minimizing the harm to the donor, and the source is more in line with ethical standards [19]. These unique advantages mean hPDLSCs are expected to become a new focus of interest in tissue regeneration engineering. Our laboratory is committed to studying hPDLSCs, using their multi-directional differentiation potential to provide new ideas for clinical treatment, and researchers around the world have also undertaken relevant studies in this area; multiple experiments have demonstrated the survival of subcutaneous hPDLSCs in the dorsal region of mice and the formation of Sharpy’s fiber-like tissue near the transplantation site [20,21]. Clinical studies have also shown that the in vivo ectopic transplantation of hPDLSCs has stable and effective advantages in the long-term follow-up of patients [22,23]. Evidence so far suggests that the use of hPDLSCs is safe and effective in the clinical repair of defects.

Therefore, in the process of skin wound healing, we tend to choose periodontal ligament stem cells derived from the oral cavity for damage repair treatment in order to obtain the curative effect of skin healing with minimal scarring. We are interested in using previously unapplied hPDLSCs for cosmetic skin repair.

It has been observed that hPDLSCs can spontaneously differentiate into MFs during in vitro amplification [24]. However, the optimum culture conditions for in vitro amplification need to be further explored. In this study, we discuss the influence of different media and cell generation on the myofibroblast differentiation ability of hPDLSCs in order to determine the optimal culture conditions and explore the myofibroblast differentiation ability of hPDLSCs in a high glucose environment by adding appropriate doses of AGEs to simulate the diabetic environment with the aim of using hPDLSCs to combat the damage caused by AGEs. In doing so, we aim to uncover the theoretical basis for the use of hPDLSCs as cellular drugs to support wound healing in diabetic patients.

## 2. Materials and Methods

### 2.1. Isolation and Cultivation of hPDLSCs

hPDLSCs were obtained from tissues attached to one-third of the tooth roots of healthy patients (5 males and 5 females), aged 15–20 years, who were undergoing orthodontic treatment at the First Affiliated Hospital of Jinan University. The isolation and culturing of hPDLSCs were performed as previously reported [25]. All experimental protocols were approved by the Ethics Committee of Jinan University (Guangdong, China) (Approval number: 2019228, 28 February 2019). The tissue was chopped, digested, and cultured in α-MEM (Gibco, New York, NY, USA) supplemented with 10% fetal bovine serum (FBS, Gibco, New York, NY, USA), 100 mg/mL streptomycin, and 100 U/mL penicillin (MDBio, Shanghai, China) at 37 °C and 5% CO_2_. The single-cell colonies were observed at passage 0 (P0). The medium was changed every 3 days. The cells from passages P0-P5 were cultured continuously to P7-P11 in α-MEM with 10% FBS medium and H-DMEM with 10% FBS medium. Cell morphology was observed under a light microscope. After fixation in 2.5% glutaraldehyde phosphate buffer overnight, the samples were dehydrated using an alcohol gradient. After drying, the samples were observed under a scanning electron microscope (SEM) and photographed.

### 2.2. MTT Assays Were Used to Detect the Inhibitory Effect of AGEs on the Proliferation of Periodontal Ligament Stem Cells

First, 3 × 10^3^ hPDLSCs/well were inoculated into 96-well plates. Each group had six repeated wells. The cells were starved for 6 h with 1% FBS medium on the second day. Six concentrations of AGEs (0, 12.5, 25, 50, 100, and 200 μg/mL, *n* = 6; Bioss, Beijing, China) were loaded in medium 24, 48, and 72 h after administration. Cell proliferation was evaluated using MTT assays. A microplate reader (Thermo Lab systems, Waltham, MA, USA) was used to detect the absorbance at 570 nm (MTT assays) after shaking the samples for 5 min. Each assay was performed in triplicate.

### 2.3. Real-Time Quantitative Fluorescent Polymerase Chain Reaction (qRT-PCR) Was Used to Detect the Expressions of α-SMA, COL1, and Vimentin

RNA was extracted according to the instructions of the Total RNA Micro Extraction Kit (Dual Column) (Tiangen Biotechnology, Beijing, China). The amount of RNA required for reverse transcription was calculated based on the measured RNA concentration, and the reaction solution was configured according to the Evo M-MLV reverse transcription instructions. qRT-PCR was performed using an SYBR-Green Quantitative PCR kit (Bio-Rad, Hercules, CA, USA). The reaction system was 20 µL. The upstream and downstream primer sequences of α-SMA (smooth muscle actin), COL-I (type I collagen), vimentin, and GAPDH (internal control) were designed by Synbiotics and synthesized by Beijing Liuhe Huada Gene Co., Ltd., Beijing, China. The primer sequences were as follows: α-SMA (F: GACAATGGCTCTGGGCTCTGTAA; R: TGTGCTTCGTCACCCACGTA); COL-I (F: AAAGATGGACTCAACGGTCTC; R: CATCGTGAGCCTTCTCTTGAG) Vimentin (F: TACGAGGAGGAGATGCGGGA; R: CATGATGTCCTCGGCCAGGT) GAPDH (F: TCGGAGTCAACGGATTTGGT; R: TTCCCGTTCTCAGCCTTGAC)

### 2.4. Western Blot Analysis Was Used to Detect Protein Expression

Cells were harvested in RIPA lysis buffer (Beyotime Institute of Biotechnology, Shanghai, China) as soon as specific experiments were complete. The protein concentration was measured using a BCA protein assay reagent kit (Thermo Scientific, Waltham, MA, USA). Proteins were separated using 12% (*w*/*v*) sodium dodecyl sulfate–polyacrylamide gel electrophoresis and transferred to a polyvinylidene fluoride membrane. After blocking with Quick Blot buffer (Applygen Technologies Inc. Beijing, China) at room temperature for 30 min, the proteins were detected after overnight incubation with anti-vimentin (1:1000 dilution, Bios, Guangzhou, China), anti-α-SMA (1:1000 dilution, BOSTER, Pleasanton, CA, USA, A03744), and anti-COL1A1 (1:1000 dilution, Cell Signaling Technology, BSN, Danvers, MA, USA, 66948). After washing, the membranes were incubated with secondary antibodies (goat anti-mouse) for 1 h at room temperature. Specific complexes were visualized using Pierce ECL chemiluminescence substrates (Thermo Scientific, Waltham, MA, USA).

### 2.5. STZ-Induced Diabetic Whole-Skin Defect Experiment to Verify Wound Healing

Eight-week-old male SPF C57BL/6J mice (*n* = 27) were purchased from Guangdong Medical Laboratory Animal Center (Guangdong Province, China). All animal experiments were approved by the Experimental Animal Management Committee of Jinan University. All mice were kept on a 12 h light/dark cycle and given regular food and water for one week. All animal experiments strictly followed the guidelines approved by the Ethical Review Committee for Animal Experiments of Jinan University (20221227-0002, 27 December 2022).

The mice were injected with streptozotocin (STZ, 50 mg/kg, Sigma Aldrich, St. Louis, MO, USA) (intraperitoneal, 50 mg/kg) and 0.1 mol/L sodium citrate buffer (pH 4.5; Solarbio, Beijing, China). One week after STZ injection, blood glucose was measured for three consecutive days, and mice with blood glucose levels >16.7 mmol/L were considered to have STZ-induced diabetes. Blood glucose testing continued until the last day of the experiment to ensure that the mice were always in a hyperglycemic state.

The diabetic mice were randomly divided into three groups: the model group (*n* = 9), vehicle group (*n* = 9), and hPDLSCs group (*n* = 9). After this, the mice were anesthetized (Sigma Corporation, Alexander City, AL, USA) (3 mL/kg). A sterile 6 mm skin punch was used to create a round, 6 mm diameter incision that penetrated the subcutaneous tissue on the backs of the mice. The blank hydrogel, which our laboratory used in previous research [26] as the vehicle, was injected as a carrier on the surface of the wound area. The hydrogel (20% Poloxamer407 + 6% Poloxamer188 + 74% H_2_O) was designed to become more gelatinous at temperatures above 37 °C, which facilitated its retention on the wound. The vehicle group underwent natural healing with the injection of the hydrogel alone, while the model group underwent natural healing with no intervention. The experimental group was administered hydrogel loaded with 50 μL hPDLSCs (1 × 10^7^). The mice were humanely sacrificed at 7, 14, and 21 days, and wound changes were measured for wound closure analysis and morphological observation. The wound area was quantified using ImageJ software 2.3.0/1.53q. Skin tissue was removed from the back after surgery, and all tissues were washed in phosphate-buffered saline, fixed in 4% formaldehyde, dehydrated in ethanol, dewaxed with xylene, and embedded in paraffin. Hematoxylin–eosin (H&E) staining, Masson staining, and immunohistochemistry (IHC) were used to detect the expression of the myofibroblast markers α-SMA and vimentin. Immunofluorescence (IF) was used to detect the expression of different types of macrophage markers, F4/80, CD206, and iNOS.

### 2.6. Immunohistochemistry (IHC) Detects α-SMA and CD31 Expression

The paraffin sections of the diabetic mice were blocked and incubated with antibodies against α-SMA (1:400 dilution, Affinity Biosciences, Cincinnati, OH, USA, BF9212) and CD31 (1:200 dilution, Affinity Biosciences, OH, USA, AF6191) overnight. After rinsing with PBS, sections were incubated with a fluorescent rabbit secondary antibody for 40 min. After washing four times with PBS, the cell nuclei were stained with DAPI, and images were acquired with a confocal laser scanning microscope (Olympus LSM 700, Carl Zeiss Microscopy GmbH, Jena, Germany).

### 2.7. Immunofluorescence (IF) Assay

After being treated, the cells on glass coverslips were fixed in 4% paraformaldehyde for 30 min, and the mouse skin was fixed in 4% paraformaldehyde for 2 days and then resected for frozen sectioning. The samples were washed three times with PBS, blocked with 10% goat serum for 30 min, and nested. Next, the slides were incubated with primary antibodies against CD206 (1:1000, Thermo Scientific, MA5-28581), iNOS (1:250, Abcam, Cambs, UK, EPR16635), and F4/80 (1:500, Abcam, Cambs, UK, EPR26545-166) overnight at 4 °C, and then the slides were incubated with anti-rabbit secondary antibody (1:10,000, Abcam, UK, Alexa Fluor^®®^ 488) in the dark at room temperature for 1 h. Finally, the sections were incubated with 4′,6-diamidino-2-phenylindole (1:10,000, DAPI, Abcam, Cambs, UK, ab104139) for 10 min and scanned using fluorescence microscopy.

### 2.8. Fluorescence In Situ Hybridization (FISH)

Cryosections were hydrated and hybridized (Advanced Cell Diagnostics, Newark, CA, USA) using the Multiplex Fluorescent Reagent v2 kit according to the manufacturer’s instructions. Nuclei were counterstained with DAPI (1:50,000, D9542, Sigma-Aldrich, DE, USA), sealed with ProLong Diamond Antifade Mountant (Thermo Fisher Scientific, P36961, MA, USA), and observed under a confocal microscope, as described above.

### 2.9. Statistical Analysis

SPSS 21.0 statistical software was used, and *p* < 0.05 was considered statistically significant. All data are shown as the mean ± standard deviation (SD). Independent samples t-tests were used to compare means between two groups. One-way analysis of variance (ANOVA) was used to determine the level of significance with GraphPad Prism 9.0 software, and *p* < 0.05 was considered to indicate statistical significance. * *p* < 0.05, ** *p* < 0.01, *** *p* < 0.001 compared to the model group.

## 3. Results

### 3.1. Spontaneous Differentiation of hPDLSCs into Myofibroblasts Was Associated with Conditioned Medium

During cultivation, both in α-MEM and H-DMEM, the hPDLSCs transformed from a fusiform shape to a flat and broad shape with the increase in cell passages. Under electron microscopy, these morphologically altered cells showed a stress-fiber-like cytoskeleton structure and extensive cytoplasm, suggesting that the hPDLSCs might differentiate into myofibroblasts (Figure 1A,B). WB analyzed the expressions of α-SMA, COL1A1, and vimentin and found that α-SMA (*p* < 0.05) and COL1A1 (*p* < 0.05) were significantly up-regulated, while vimentin was decreased (*p* < 0.05) in H-DMEM. In α-MEM, with the increase in cell generation, α-SMA increased prominently, as it did in α-MEM (*p* < 0.05), but COL1A1 and vimentin expression showed no significant difference (Figure 1C).

At the level of the gene, α-SMA was increased in H-DMEM, while the expression change of COL1 was not significant, and vimentin was significantly down-regulated (*p* < 0.001) (Figure 1D). There was no significant difference in the relative gene expression of α-SMA and vimentin in α-MEM, but the relative gene expression of COL1 increased (*p* < 0.001). Overall, α-MEM was slightly better than H-DMEM, but there was no significant difference between the two groups (*p* > 0.05). Although α-MEM was slightly better for the multi-directional differentiation of hPDLSCs, the difference between the two media was not significant. H-DMEM was sufficient to simulate the effects of a hyperglycemic environment on the cells in later experiments. The same batch of hPDLSCs which used in experiments have been authenticated by FCM (Appendix A) and confirmed that PDLSCs have the ability of Osteogenesis and adipogenic differentiation (Appendix A).

### 3.2. hPDLSCs Resist the Damage Caused by AGEs

After treatment with different concentrations of AGEs for 24, 48, and 72 h, MTT assays showed that the proliferation ability of hPDLSCs decreased gradually with the increase in AGE concentrations and time. The IC50 values were 457.6, 85.4, and 49.6 μg/mL at 24, 48, and 72 h. Therefore, 48 h was chosen as the administration time for the subsequent experiments. In order to avoid the excessive inhibitory effect of AGEs on the proliferation of hPDLSCs and the interference of subsequent myofibroblastic differentiation experiments, the concentration of AGEs was selected as 25 μg/mL (Figure 2A). The same MTT assays were used to evaluate the effect of AGEs at the concentration of 25 μg/mL on the growth of hPDLSCs at the seventh passage (P7) compared with P11. When treated with 25 μg/mL AGEs for 48 h, the relative proliferation rate of the P11 cells decreased more significantly than that of the P7 cells (*p* < 0.05), so P7 was selected for subsequent cell experiments (Figure 2B). Compared with the control group, the relative gene expression of α-SMA in the AGE group was significantly decreased on the 4th and 8th day of treatment (*p* < 0.001). Compared with the 4th day of treatment, α-SMA was up-regulated in the AGE group after the 8th day of treatment (*p* < 0.05), which indicated that the AGEs only inhibited the expression of α-SMA in hPDLSCs to a certain extent; the hPDLSCs still had the ability to express α-SMA in an environment that included AGEs (Figure 2C). According to WB, the expression levels of α-SMA, COL1A1, and vimentin protein in the hPDLSCs were significantly affected by AGEs at day 4 and day 8. Compared with the control group, the protein expression of α-SMA in the AGE group did not change significantly on day 4 and decreased slightly on day 8, but the difference was not statistically significant (*p* > 0.05). Compared with the control group, the expression of COL1A1 in the AGE group was significantly down-regulated on the 4th and 8th day of treatment (*p* < 0.01). Compared with the control group, the expression of vimentin in the AGE group was significantly down-regulated on the 4th day, and then on the 8th day, the difference was no longer statistically significant (*p* > 0.05) (Figure 2D–F).

The observation of cell morphology using electron microscopy and scanning electron microscopy was carried out on the 2nd, 4th, 6th, and 8th day after AGE induction. Freshly inoculated primary cells were suspended in culture flasks, and after 24 h, most of the cells adhered to the wall, with multiple short protrusions forming in the cell membrane. Two days later, some of the cells extended into a spindle shape, and the processes fused and disappeared. Later, the cells were completely extended, showing a long spindle shape with hypertrophy of the cell and gradually narrowing at both ends. After 4–8 days, 80% confluence was achieved, and the cells changed to a more flat and broad morphology. However, compared with the model group, the fiber-like structure and the number of cells in the induced group were significantly reduced (Figure 2G). The SEM results indicated that AGEs cause damage to the morphology and structure of PDLSCs (Figure 2H).

These results indicated that AGEs inhibit the expression of α-SMA, COL1, and vimentin, and AGEs can impair the myofibroblast differentiation ability of hPDLSCs.

### 3.3. hPDLSCs Repaired Full-Thickness Skin Defects in Diabetic Mice

Our in vitro studies led us to evaluate the wound healing capacity of hPDLSCs using an excisional skin wound animal model. The hPDLSCs were used to cover the skin defects of diabetic mice. The wound condition was observed, and the wound healing was evaluated on days 0, 3, 7, 11, 14, and 21 after the operation. The tissues were then stained using H&E, Masson, IHC, and IF (Figure 3D). All wounds showed slow recovery and healing without any signs of infection (Figure 3A). Three days after surgery, the dorsal skin wounds of the three groups of mice began to shrink, without local edema and exudation, and the wounds in the positive group contracted obviously. At day 7 (D7), the wound had formed a hard crust that was dry and without exudation. At D14, the wounds in the hPDLSCs group and the vehicle group were almost completely healed, while those in the model group had a small range of pink, soft crusts on the surface. At D21, the wounds in the three groups were almost invisible and covered with hair, suggesting that all groups were close to complete healing. The wound healing trajectory map also confirmed this phenomenon (Figure 3B). The quantitative analysis of wound closure at different time points showed that the wound healing rate of the hPDLSCs group was significantly higher than that of the model group at D3, D7, and D11 (Figure 3C). Compared with the model group and vehicle group, hPDLSCs promoted healing at a faster rate, indicating that hPDLSCs may resist the negative effects of AGE injury and accelerate wound healing.

Next, we performed histological analyses to evaluate re-epithelialization and wound healing properties. H&E staining confirmed that hPDLSCs accelerated wound closure and healing (Figure 4). Histological analysis of wounds at different time intervals (D7, D14, and D21) showed that, compared with the model group, the hPDLSCs-treated animals had more granulation tissue and thick and dense collagen fibers in the wounds. In the hPDLSCs and positive group, the new epidermis was closely combined with the dermis, the skin layers were obvious, and the tissue structure was relatively complete. At D21, fibroblasts parallel to the epidermis could be seen in the superficial layer of the dermis in the hPDLSCs group. The cross-sectional area of the wound in the hPDLSCs group was smaller than that in the model group at D14 and D21, and the collagen content in the vehicle group and hPDLSCs group was higher than that in the model group. These findings all confirmed better wound healing in the presence of hPDLSCs. Masson staining showed that on the 14th day after modeling, collagen fibers began to repair in the hPDLSCs group and the positive group, while there was less collagen deposition in the model group, and it was only deposited in the dermis and less so in the basal part. On the 21st day, the collagen formation in the hPDLSCs group was thicker and more regularly arranged than that in the model group and the vehicle group.

The results of the immunohistochemical staining of α-SMA and COL1A1 in the skin tissue sections at D14 and D21 after the operation are shown in Figure 5. At D14, there was only a small amount of positive expression in the wound skin of the model group, while the integrated option density (IOD) value of α-SMA in the hPDLSCs group and the positive group was significantly up-regulated (*p* < 0.05, vs. model). The expression of COL1A1 was significantly positive (*p* < 0.01). At D21 after the operation, α-SMA in the treatment group (hPDLSCs group and vehicle group) was significantly decreased compared with the model group (*p* < 0.05). CD31 immunohistochemical staining was used to observe the skin neovascularization at 21 days after the operation. Only a small degree of CD31 expression was observed in the model group, and the positive expression in the hPDLSCs group was significantly increased (*p* < 0.01 vs. model). Compared with the hPDLSCs group, the IOD value of the positive group was higher, and the difference was statistically significant (*p* < 0.01). This suggests that hPDLSCs are equipped to promote angiogenesis.

Furthermore, the markers of the M1 and M2 macrophages, iNOS and CD206, were observed in the mouse skin tissues from immunofluorescence and confocal microscopy imaging. The results showed that compared with the model and vehicle groups, the CD206 immunofluorescence co-localization in the stem cell treatment group was higher, indicating that the expression of M2 anti-inflammatory macrophages was higher, with a corresponding anti-inflammatory effect. In contrast, M1 was lower in the mice treated with hPDLSCs than in the other groups (Figure 6). These results indicate that hPDLSCs not only promote wound healing but also promote the transformation of macrophages into the M2 anti-inflammatory type, which has a significant anti-inflammatory effect during the wound healing process. Periodontal Ligament-associated Protein 1 (PLAP-1) can be considered as a specific extracellular matrix protein to label the lineage of periodontal ligaments, including stem/progenitor cells. Therefore, we performed FISH to detect the expression of hPDLSCs in the skin of the mice to evaluate the effect of externally injected hPDLSCs on skin wound healing. The injection of hPDLSCs are expressed in the skin of mice to PLAP-1 positive cells (Figure 7A), but no PLAP-1 expression in the control group (Figure 7B). These results indicate that hPDLSCs could survive in diabetic mice and differentiate into cells in the skin.

## 4. Discussion

Stem cells have become a research hotspot in tissue engineering in recent years. Stem cells usually come from bone marrow [25]. Because of their self-renewal properties, plasticity of differentiation potential, and immunosuppressive and anti-inflammatory functions, stem cells have gradually become the key to cell therapy and are widely applied in regenerative medicine. Stem cell therapy is also becoming an emerging beauty trend. However, the difficulty of obtaining stem cells limits their research applications. Therefore, it is essential to seek adequate ways to obtain substantial stem cell sources.

The first batch of hPDLSCs was isolated from in vitro tissues by cloning technology [26]. Studies have found that hPDLSCs have strong survival, proliferation, and differentiation abilities in vitro, and it is also possible to stimulate their regenerative potential after transplantation in vivo [13]. It was found that grafted hPDLSCs can produce connective tissue-like structures, bone, and collagen fibers in the periodontal ligament without adverse effects [27]. Park et al. transplanted hPDLSCs into the backs of immunodeficient mice to form complexes with the most commonly used calcium phosphate scaffolds, HA and beta-tricalcium phosphate (HA/TCP), and found that they expressed periodontal tissue-like structures in the subcutaneous tissues [13]. Feng et al. reported that autologous human hPDLSCs could achieve a stable regeneration effect in vivo [28]. These experiments showed that hPDLSCs, as an alternative therapy, can be used to repair defects with stable effects and few ethical complications.

MFs play a critical role in both the proliferative and remodeling stages of wounds. MFs regenerate connective tissue by secreting and organizing extracellular matrix components, including different types of collagen, proteoglycans, and signaling molecules [29]. Therefore, the origin, activity, apoptosis, and functioning of MFs may affect the wound-healing process [9]. Whether hPDLSCs can perform myofibroblast differentiation determines whether they can be applied in wound healing and repair. In previous studies, α-SMA has been the most commonly used molecular marker of MFs, but this molecular marker is also highly expressed in other cells, such as smooth muscle cells. Thus, COL1A1 was also added in this study as an additional differentiation indicator to demonstrate the transformation of hPDLSCs into myofibroblasts. In addition to hPDLSCs, an increase in COL1A1 can also be observed when other cells differentiate into MFs, which has significance for the formation of the extracellular matrix in wound healing [30]. Vimentin, a cellular marker of epithelial–mesenchymal transition (EMT), is a common component of wound healing and is therefore included in the measurement.

Generally, the recognized cell generation used for experiments has been P3-7. Based on previous experimental experience, this study extracted hPDLSCs and cultured them to P7-11 for the experiments. The hPDLSCs changed from a fusiform to MF-like stress-stretching morphology, accompanied by increased levels of markers related to myofibroblastic differentiation. Our experiment showed that the cellular morphology changed significantly from P9, which was consistent with previous research results [24]. With the increase in cell generation in H-DMEM, the expression of α-SMA and COL1A1 was increased, indicating that the hPDLSCs showed a trend of myofibroblastic differentiation during the in vitro proliferation process.

On this basis, we found that hPDLSCs using α-MEM had slightly better differentiation ability than H-DMEM, but the difference was not significant (*p* > 0.05). The comparison of the media showed that the α-MEM medium supplemented more amino acids, ascorbic acid, biotin, ribonucleosides, deoxyribonucleosides, and vitamin B12 than H-DMEM. These components are important factors in cell survival, clonal growth, energy metabolism, and cell protection [31,32], which may explain the results. In view of the fact that this experiment aimed to explore the function of hPDLSCs in wound healing, and as myofibroblastic differentiation will be discussed later, here, we discuss the use of H-DMEM to verify whether hPDLSCs can resist injury from AGEs. It is worth mentioning that if the multi-differentiation potential of hPDLSCs is needed for wound repair, α-MEM should be selected to achieve better fiber and bone tissue repair.

The changes in reactive oxygen species (ROS) and hypoxia in adjacent cells during high glucose levels are the main reasons for the changes in stem cell signaling. The production and persistence of ROS will severely inhibit antioxidant enzymes and non-enzymatic antioxidants in various tissues, further aggravating oxidative stress [33], inhibiting the proliferation of stem cells, and slowing down tissue repair and healing, as well as damaging innate tissue repair and regeneration [4]. Previous studies have shown that diabetes can cause the dysfunction of mesenchymal stem cells in vivo by affecting cell vitality, proliferative ability, stemness, and pluripotency [34,35]. Hyperglycemia and an inflammatory environment will damage the survival and secretion of mesenchymal stem cells [36], which can cause difficulties in treatment. Therefore, it is necessary to explore whether the AGE microenvironment affects the spontaneous myofibroblastic differentiation ability, which is related to whether hPDLSCs can repair diabetic wounds efficiently.

In our experiments, we found that the proliferation ability of PDLSCs gradually decreased with the increase in the AGE concentration, suggesting that the proliferation ability of stem cells was strongly affected by high levels of glucose. To further verify the effect of AGEs on PDLSCs, we found that the expression of the myofibroblast surface markers α-SMA and vimentin was decreased after AGE injury. After intervention with AGEs for 4 d and 8 d, we found that the expression of COL1A1 in the hPDLSCs decreased, indicating that AGEs caused damage to the collagen synthesis ability of hPDLSCs. This indicated the effectiveness of the model. The activation of the classical Wnt/β-catenin pathway is one of the mechanisms by which AGEs decrease the level of COLIA1 [37]. The WB results showed that the protein expression of α-SMA was slightly decreased after treatment with AGEs, but the difference was not statistically significant. Moreover, the inhibitory effect of AGEs on vimentin was gradually reduced with the increase in generations. These results indicate that hPDLSCs may have some ability to resist the damage caused by AGEs, and their ability to differentiate into myofibroblasts is not significantly inhibited, which further indicates that hPDLSCs have potential in diabetic dermal wound healing.

AGE deposition in a high glucose environment causes wound healing difficulties in diabetic patients. This is also a challenge for cosmetic skin restoration. Common treatment strategies for curing diabetic wounds include debridement, infection control, lower limb revascularization, decompression, and ultrasound [38]; these strategies can accelerate wound healing. However, the currently available treatment methods are still limited in terms of their therapeutic effects, especially for oral inflammation healing in diabetic patients. Therefore, we have begun to pay attention to periodontal ligament stem cells that have not yet become therapeutic cells and the application of such stem cells in chronic wound healing. Studies have found that animals with diabetes show higher levels of inflammation, more apoptotic cells, and less fibroblast proliferation [39]. Additionally, epithelial and connective tissue repair is delayed by increased apoptosis and decreased cell proliferation [40]. Therefore, the hPDLSCs selected in our study are worthy of further study for their various changes and tolerance to a high glucose environment in vivo.

The skin wound model of diabetic mice was established, and the cultured hPDLSCs were injected into five points of the margin of the wound bed. The effectiveness of hPDLSC transplantation was evaluated using the wound closure area, inflammatory infiltration, collagen deposition, and arrangement. It should be noted that through a previous study, we found that the survival rate of P7 cells was higher than that of P11 cells under the intervention of hPDLSCs in H-DMEM medium at the same concentration of AGEs. This indicated that, although P11 had a higher myofibroblastic differentiation ability compared with P7, P11 had a lower survival ability in the diabetic microenvironment. Thus, P7 cells were selected for subsequent animal experiments.

Our results showed that a circular wound was established on the skin of STZ-induced diabetic mice after successful modeling, and the wound was obvious on the 3rd day after surgery, with no redness or swelling of the surrounding mucosa. It is worth noting that the appendages (hair follicles, etc.) of the dermis after diabetic wound healing were different from those of normal mice. However, according to the comparison of the width of the wound bed in each figure, it was found that the width of the wound bed in the hPDLSC treatment group was significantly smaller than that in the vehicle group and the model group, indicating that hPDLSCs have the potential to repair the dermis and, thus, achieve the closure of the wound.

From the perspective of wound closure speed, the healing speed of the hPDLSCs group was significantly faster than that of the model group. The HE results showed that 14 days after surgery, compared with the hPDLSCs group, there was still more inflammatory cell infiltration in the model group, which suggested that the inflammatory period in the wound healing process of the model group might be prolonged; that is, the proliferation period started relatively late, which seemed to be related to the epidermis and dermis separation. At day 21, more fibroblasts parallel to the epidermis were observed in the dermis of the hPDLSCs group compared with those in the model group. We suggest that this might have been related to the paracrine mechanism of hPDLSCs [41]. After injection into the wound, hPDLSCs can release cytokines, such as TGF-β1, that stimulate fibroblast proliferation [42]. Masson staining showed that the collagen density in the hPDLSCs group was higher than that in the model group at day 14, and the collagen arrangement was more orderly, with abundant blood vessels in the base. This indicated that, although there was no significant difference in the wound size between the model group and the hPDLSCs group at day 14 on the surface, the hPDLSCs improved the healing of the dermis and basal tissues compared to the model group.

It is worth noting that our experiment found that the α-SMA content was the highest in STZ-induced diabetic mice on day 14. On the 21st day after surgery, IHC showed that the expression of α-SMA in the hPDLSCs group was decreased, while it was still increased in the model group. This suggests a delay in myofibroblast differentiation during the early healing stage of diabetes, in congruence with the conclusions of Retamal et al. [43]. The mechanism of promoting healing may be related to the appearance of α-SMA, as MFs do not persist, and while they are highly expressed in α-SMA, their differentiation is delayed in diabetic wounds. With a prolonged healing time, α-SMA is still highly expressed at a later time point. In normal wounds, MFs appear early in healing and are reduced during tissue remodeling, but MF activity should be reduced or eliminated by apoptosis during late healing [44].

## 5. Conclusions

In summary, our study reveals important changes in the differentiation of periodontal ligament stem cells into myofibroblasts in diabetic wounds, as well as the ability of hPDLSCs to resist damage from AGEs in vitro and in vivo. The spontaneous differentiation ability of hPDLSCs and their ability to resist injury from AGEs means they can be used to repair wounds, which provides a theoretical basis for the use of stem cell therapy in wound healing and repair in clinical diabetic patients. This process is expected to be applied in cosmetic dermatology in the future.

## Figures and Tables

**Figure 1 bioengineering-11-00602-f001:**
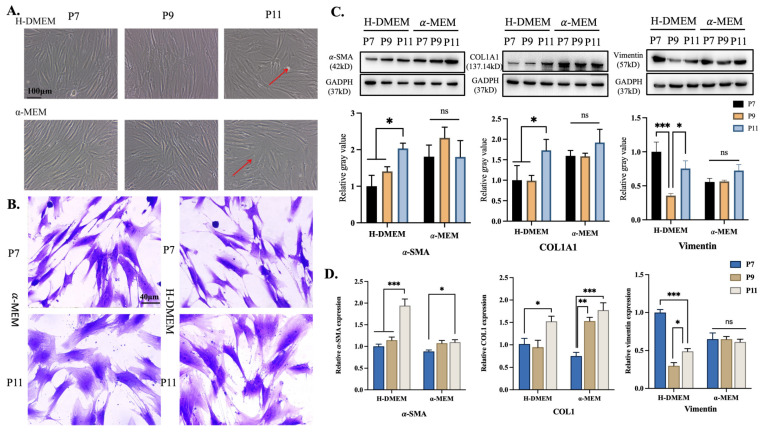
Characteristics of fibroblast differentiation of hPDLSCs in vitro. (**A**) Spontaneous differentiation of periodontal ligament stem cells was observed in H-DMEM and α-MEM medium. During in vitro culture, periodontal ligament stem cells changed from a spindle-like shape to a flat, broad shape with the increase in cell generations. Red arrows represent the shape-changed cells.(**B**) The 7th and 11th generation hPDLSCs were selected and stained with crystal violet. Under electron microscope, P7 showed a typical spindle morphology, and in the latter, the morphology tended to be the flat morphology of myofibroblasts. (**C**) Expression and semi-quantitative analysis of the three proteins in different generations and in different mediums detected by WB. (**D**) qPCR was used to detect the expression levels of myofibroblast differentiation-related genes (H-DMEM, α-MEM, hPDLSCs, COL1A1: type 1 collagen α1 chain; α-SMA, vimentin, *n* = 3, ns *p* > 0.05, * *p* < 0.05, ** *p* < 0.01, *** *p* < 0.001).

**Figure 2 bioengineering-11-00602-f002:**
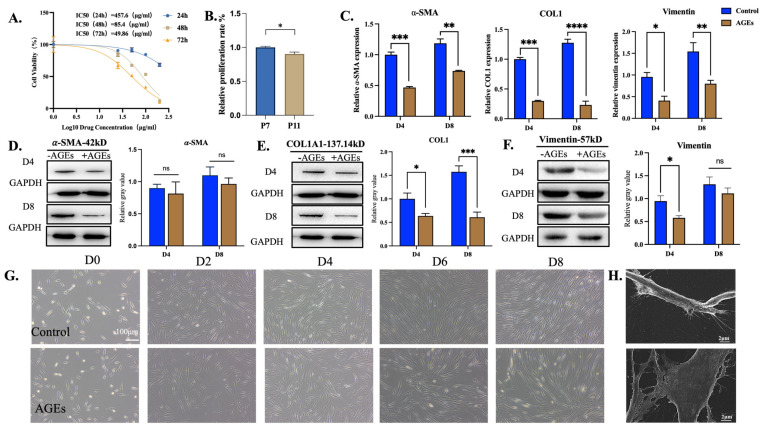
Effects of AGEs on the fibroblast differentiation ability of hPDLSCs. (**A**) Cell viability of hPDLSCs (P7) cultured with different concentrations of AGEs for 24 h, 48 h, and 72 h. The IC50 was 457.6 μg/mL, 85.4 μg/mL, and 49.6 μg/mL at 24 h, 48 h, and 72 h, respectively. (**B**) MTT was used to detect the effect of 25 μg/mL AGEs on the proliferation of hPDLSCs at different generations at 48 h. (**C**) Effect of AGEs culture environment on the relative gene expression of α-SMA, COL1A1, and vimentin in hPDLSCs at day 4 and day 8 detected by qPCR. (**D**) Effect of AGEs culture environment on the protein expression levels of α-SMA in hPDLSCs at day 4 and day 8 detected by WB and their semi-quantitative analysis. (**E**) Effect of AGEs culture environment on the relative gene expression of COL1 in hPDLSCs at day 4 and day 8 detected by WB and their semi-quantitative analysis. (**F**) Effect of AGEs culture environment on the relative gene expression of vimentin in hPDLSCs at day 4 and day 8 detected by WB and their semi-quantitative analysis. (**G**) AGEs can inhibit the proliferation of hPDLSCs. At the time of inoculation, the cells presented a round suspension, which had been attached to the wall and formed a spindle shape by day 2, and the whole medium was spread after 4–8 days. (**H**) The morphology of hPDLSCs in the AGEs group was significantly different from the control group by electron microscopy (*n* = 3, ns *p* > 0.05*, * p* < 0.05, ** *p* < 0.01, *** *p* < 0.001).

**Figure 3 bioengineering-11-00602-f003:**
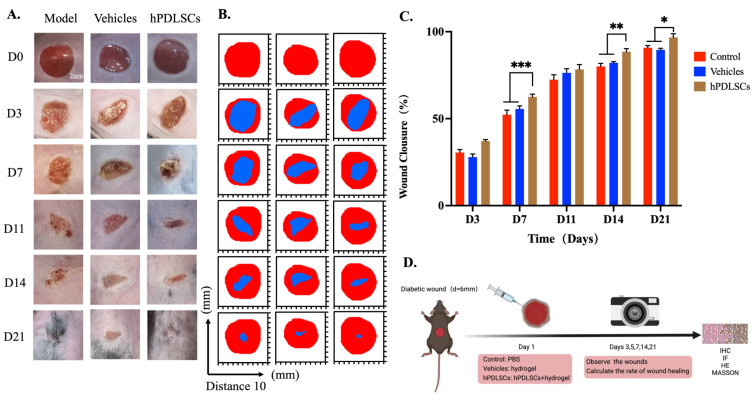
Treatment of diabetic wounds in mice with hPDLSCs transplantation. (**A**) Process observation of wound healing. On the 3rd day (D3), the back skin wounds of the two groups began to shrink, and a hard scab was formed on the D7 in the hPDLSCs group. On D14, both the control group and the hPDLSCs group were healed obviously, and a small range of pink, soft scabs was left on the surface of the control group. On D21, the wounds were almost invisible and covered with hair, suggesting that the wounds were nearly completely healed. (**B**) Schematic of the wound healing trajectory. With the increase in days, the skin wounds of the two groups decreased. (**C**) Quantitative analysis of wound closure at different time points. (**D**) Schematic diagram of a mice wound model after intraperitoneal injection of low-dose streptozotocin and sodium citrate buffer (*n* = 3, * *p* < 0.05, ** *p* < 0.01, *** *p* < 0.001).

**Figure 4 bioengineering-11-00602-f004:**
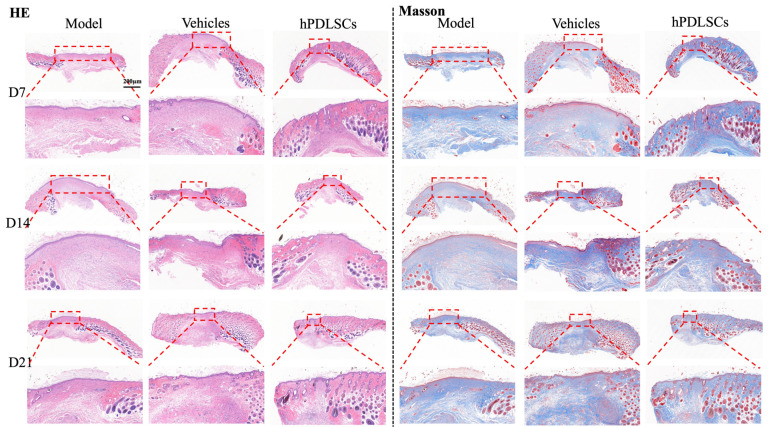
Diagrams of the sections stained with H&E and Masson at D7, D14, and D21 after surgery under the 4- and 10-fold electron microscope. H&E staining showed that the new epidermis in the hPDLSCs group was tightly combined with the dermis, indicating that the hPDLSCs group significantly promoted re-epithelialization of the wound. Inflammatory cell infiltration was observed in all three groups at 14 days after operation, with the most obvious in the control group and the least in the periodontal ligament stem cell group. The red dotted box represents the wound bed, and the hPDLSCs group had the smallest wound cross-sectional area on D14 and D21. The Masson staining plot showed that at D14, compared with the hPDLSCs group, the other two groups had less collagen deposition, which was only deposited in the dermis. At D21, the collagen arrangement in the hPDLSCs group was more uniform and tight, the wound bed was narrowed and had almost disappeared, and the skin appendage was almost completely recovered, suggesting that hPDLSCs may have the ability to make diabetic wounds close to ordinary tissue healing.

**Figure 5 bioengineering-11-00602-f005:**
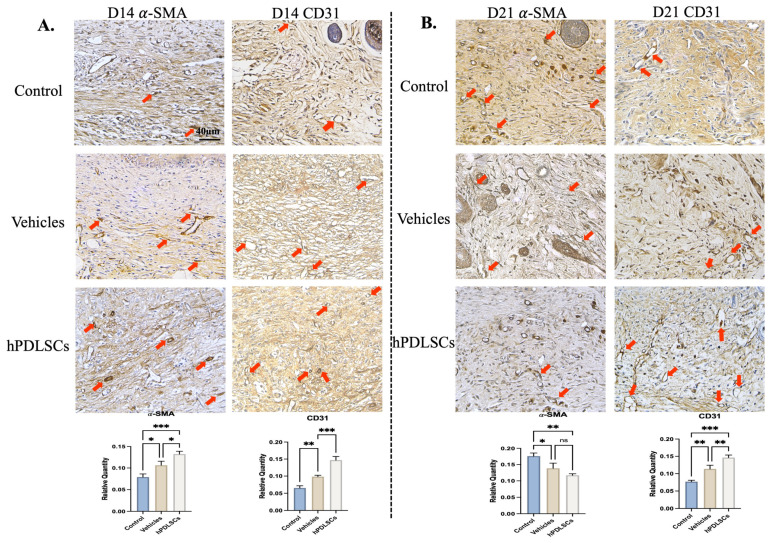
Immunohistochemical detect antigen expression of α-SMA, CD31 at D14 and D21. (**A**) The expression of α-SMA and CD31 in the stem cells treatment group and blank control group on the 14th day and the semi-quantitative analysis of them. (**B**) The expression of α-SMA and CD31 in the stem cells treatment group and blank control group on the 21st day and the semi-quantitative analysis of them. The red arrows represent positive expression areas (*n* = 3, * *p* < 0.05, ** *p* < 0.01, *** *p* < 0.001).

**Figure 6 bioengineering-11-00602-f006:**
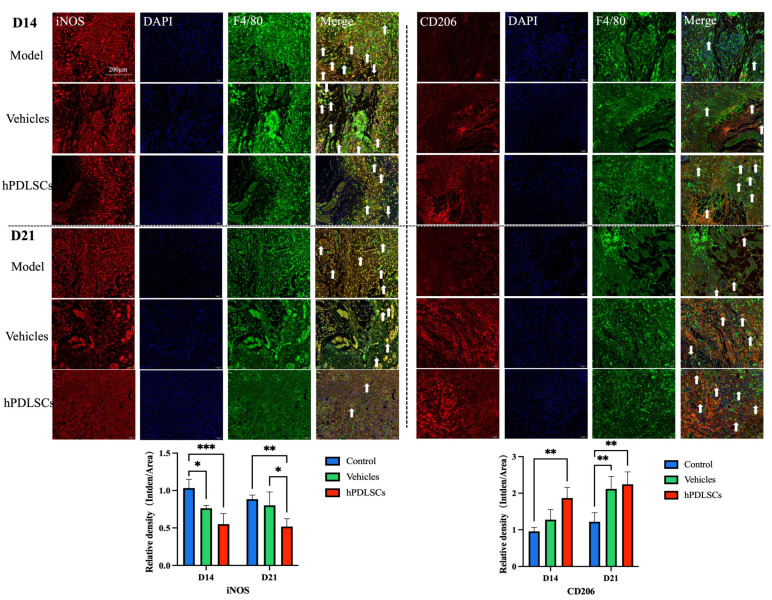
Immunofluorescence staining of iNOS, CD206, and F4/80 in wound tissue at D14 and D21. The red channel represents iNOS or CD206, the blue channel represents DAPI, and the green channel represents F4/80. The positive areas have been pointed out using white arrows. In this figure, iNOS indicates the type1 macrophage (M1), while CD206 indicates the type2 macrophage (M2). The expression of CD206 in the hPDLSCs group is significantly higher than the other groups, while the expression of iNOS in the hPDLSCs group is significantly lower than the other groups (*n* = 3, * *p* < 0.05, ** *p* < 0.01, *** *p* < 0.001).

**Figure 7 bioengineering-11-00602-f007:**
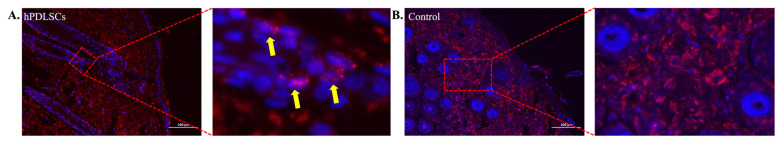
Fluorescence in situ hybridization of PLAP-1. The red channel represents PLAP-1, the blue channel represents DAPI, and the yellow arrows represent PLAP-1 positive cells. There are several PLAP-1 positive cells in the hPDLSCs group, while none in the control group.

## Data Availability

The authors confirm that the data supporting the findings of this study are available within the article and its Appendix A.

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
