# Peer review of "Human Periodontal Ligament Stem Cells (hPDLSCs) Spontaneously Differentiate into Myofibroblasts to Repair Diabetic Wounds"

_bioengineering, 2024, doi:10.3390/bioengineering11060602_

Round 1

Reviewer 1 Report

Comments and Suggestions for Authors

Mesenchymal stem cells (MSCs) from various tissue sources are currently considered a promising tool for regenerative medicine and cell therapy. PDLSCs are in high demand as cell products for regenerative medicine applications.

 The title needs to be revised to better reflect the data on MS

 The comparison between α-MEM and H-DMEM. It is unclear why the authors chose these particular formulations.

The protocol for electron microscopy is not presented in the Materials and Methods section.

L183 “…expression of myofibroblast marker genes: α-SMA and   vimentin” The Reviewer is assuming that this statement is not correct. Firstly, how   could immunocytochemistry be applied for genes evaluation.  Secondly,   α-SMA and   vimentin  antigens could not   be considered as myofibroblast markers only. These proteins are well represented in other stromal lineage cells like smooth muscle cells for instance.

 L184 Similarly,  “Immunofluorescence (IF) were used to detect the expression of different type  macrophage marker genes: F4/80, CD206 and iNOS”. IF for gene, how is it  possible?

L215    Flattening, increase in size, and formation of stress  fibers  along sub-cultivations could be considered as signs of replicative  senescence rather than spontaneous differentiation into myofibroblasts. The identification of genes and molecules related to mesenchymal cells is not sufficient evidence of myofibroblast commitment.  Have the authors performed any functional tests, such as collagenous gel contraction?

It is generally accepted that exogenous multipotent mesenchymal stem/stromal cells (MSCs), to which PDLSCs belong, are retained in the body for a short time before being eliminated. The potential of MSCs for repair through direct differentiation into myofibroblasts   is a matter of speculation. Therefore, it is important to distinguish between in vitro studies and in vivo differentiation observations when discussing the contribution of MSCs or PDLSCs in wound healing.

The MS exhibits poor writing quality. The Introduction and Discussion sections are extremely stretched and vague. These sections should be completely revised.  

Extensive editing of English language required

Author Response

1.Mesenchymal stem cells (MSCs) from various tissue sources are currently considered a promising tool for regenerative medicine and cell therapy. PDLSCs are in high demand as cell products for regenerative medicine applications.

The title needs to be revised to better reflect the data on MS

The comparison between α-MEM and H-DMEM. It is unclear why the authors chose these particular formulations.

Response: Thank you for your comment.

Title:Human Periodontal Ligament Stem Cells (hPDLSCs) Spontaneously Differentiate into Myofibroblasts to Repair Diabetic Wounds

We referred to the literature of Kengo Iwasaki[1], who used α-MEM containing 10% FBS for their experiments. The media commonly used in cell experiments include BME, MEM, DMEM, and IMDM[2]. H-DMEM has a higher glucose concentration than α-MEM. Therefore, we chose the most common H-DMEM to compare with α-MEM.

References:

  • Iwasaki K, Komaki M, Akazawa K, Nagata M, Yokoyama N, Watabe T, Morita I. Spontaneous differentiation of periodontal ligament stem cells into myofibroblast during ex vivo expansion. J Cell Physiol. 2019 Nov;234(11):20377-20391.
  • Kong H, Wang L, Zhu Y, Huang Q, Fan C. Culture medium-associated physicochemical insights on the cytotoxicity of carbon nanomaterials. Chem Res Toxicol. 2015 Mar 16;28(3):290-5.

2.The protocol for electron microscopy is not presented in the Materials and Methods section.

Response: We are grateful for this suggestion. After paragraph 120, we added “After fixation in 2.5% glutaraldehyde phosphate buffer overnight, the samples were dehydrated using an alcohol gradient. After drying, the samples were observed under scanning electron microscopy (SEM) and photographed.”

3.L183 “…expression of myofibroblast marker genes: α-SMA and   vimentin” The Reviewer is assuming that this statement is not correct. Firstly, how   could immunocytochemistry be applied for genes evaluation.  Secondly,  α-SMA and   vimentin  antigens could not   be considered as myofibroblast markers only. These proteins are well represented in other stromal lineage cells like smooth muscle cells for instance.

Response: We are grateful for this suggestion. Firstly, the relevant term has been corrected from "marker gene" to "marker". Secondly, in vitro, the hPDLSCs were the only cell type in the pure cell culture. The cell types were determined by relevant tests and morphological observation. As a result, these two markers could be used to identify MFs in cell experiments. In in vivo experiments, smooth muscle cells mainly exist in the muscle tissue, and the muscle layer is located under the subcutaneous tissue layer. However, the tissue harvested in our experiment was the dermis layer, allowing us to avoid the muscle tissue and ensure that there were no smooth muscle cells or other stromal lineage cells.

4.L184 Similarly, “Immunofluorescence (IF) were used to detect the expression of different type  macrophage marker genes: F4/80, CD206 and iNOS”. IF for gene, how is it  possible?

Response: Thank you for your comment. We are sorry for this oversight. In paragraph 191 in the updated version, the phrase has been corrected from “marker genes” to “marker”.

5.L215    Flattening, increase in size, and formation of stress  fibers  along sub-cultivations could be considered as signs of replicative  senescence rather than spontaneous differentiation into myofibroblasts. The identification of genes and molecules related to mesenchymal cells is not sufficient evidence of myofibroblast commitment.  Have the authors performed any functional tests, such as collagenous gel contraction?

Response: Thank you for your comment. It has been documented that stem cells corresponding to passages 11-14 show adequate cell function[1]. The stem cells selected for our experiments were of a more viable generation, with the progress of replication still in a relatively vigorous stage; they had not entered the senile stage. The corresponding verification experiments will be carried out in future research.

References:

[1] Martin-Piedra MA, Garzon I, Oliveira AC, Alfonso-Rodriguez CA, Carriel V, Scionti G, Alaminos M. Cell viability and proliferation capability of long-term human dental pulp stem cell cultures. Cytotherapy. 2014 Feb;16(2):266-77.

6.It is generally accepted that exogenous multipotent mesenchymal stem/stromal cells (MSCs), to which PDLSCs belong, are retained in the body for a short time before being eliminated. The potential of MSCs for repair through direct differentiation into myofibroblasts   is a matter of speculation. Therefore, it is important to distinguish between in vitro studies and in vivo differentiation observations when discussing the contribution of MSCs or PDLSCs in wound healing.

Response: Thank you for your comment. hPDLSCs with myofibroblastic differentiation were cultured in H-DMEM medium in vitro, and the 9th passage of differentiated cells were injected into the wound area of STZ rats to observe the effect on wound healing.

7.The MS exhibits poor writing quality. The Introduction and Discussion sections are extremely stretched and vague. These sections should be completely revised. Extensive editing of English language required.

Response: We are grateful for this suggestion. In terms of language, we have carried out professional language touch-ups and revised the language of the whole paper. The English proof-reading certification (verification code: english-78466 ) is included in the attachment. 

Reviewer 2 Report

Comments and Suggestions for Authors

The paper by Li and coworkers discussed that human periodontal ligament stem cells (hPDLSCs) would spontaneously differentiate into myofibroblasts in a specific culture, which makes them treat diabetic wounds. This is an interesting manuscript, but there are places where more complete information is needed.

1. In the section "2.3. Real-Time Quantitative Fluorescent Polymerase Chain Reaction (qRT-PCR)," it is mentioned that the primers for α-SMA, COL1, and Vimentin were designed and synthesized, but the primer sequences for COL-III were not provided, and Vimentin primers were mentioned instead. Additionally, COL-III results were not reported in the entire manuscript.

2. In section "2.5. STZ-Induced Diabetic Whole Skin Defect Experiment to Verify Wound Healing," there is a discrepancy in the concentration of streptozotocin (STZ) mentioned, with 50 mg/kg in the text and 45 mg/kg in parentheses. Clarification is needed on the correct concentration. Furthermore, a brief description of the hydrogel used in the experiment, which was mentioned as a carrier on the wound surface, is requested.

3. In the "3.1 Spontaneous Differentiation of hPDLSCs into Myofibroblasts" section, clarification is sought regarding the rationale for measuring vimentin expression. Is vimentin upregulated or downregulated during myofibroblast differentiation? In Figure 1C, which represents western blotting, the Y-axis in the quantitative bar graph should be corrected to indicate relative protein expression for all proteins, not just COL1A1.

4. In Figure 2D, the western blotting results show a subtle decrease in α-SMA with AGEs D8 treatment, but the quantitative bar graph does not fully support this observation. Additional western blotting images with corresponding data are requested for validation. Additionally, an explanation is needed for the absence of vimentin analysis in the cellular experiments involving AGEs in this section.

 5. If the statement "At D21, the wounds in the three groups were almost invisible and covered with hair, suggesting that all groups were close to complete healing" is accurate, then the bar graph in Figure 3C for D21 should approach 100%. Moreover, the healing percentages for D14 and D21 in Figure 3C are similar across all groups, around 80%, regardless of the treatment.

 6. Regarding Figure 5, an explanation is sought for why α-SMA is highest in the hPDLSCs group at D14 but lowest at D21.

 7. Figure 6 lacks legends; kindly provide the necessary legends for clarity.

Author Response

Reviewer #2:

1.In the section "2.3. Real-Time Quantitative Fluorescent Polymerase Chain Reaction (qRT-PCR)," it is mentioned that the primers for α-SMA, COL1, and Vimentin were designed and synthesized, but the primer sequences for COL-III were not provided, and Vimentin primers were mentioned instead. Additionally, COL-III results were not reported in the entire manuscript.

Response: Thank you for your comment. We are sorry for this oversight. Contrary to what was originally stated in paragraph 141, we did not perform experiments related to COL-III, so the information about COL-III was removed.

2.In section "2.5. STZ-Induced Diabetic Whole Skin Defect Experiment to Verify Wound Healing," there is a discrepancy in the concentration of streptozotocin (STZ) mentioned, with 50 mg/kg in the text and 45 mg/kg in parentheses. Clarification is needed on the correct concentration. Furthermore, a brief description of the hydrogel used in the experiment, which was mentioned as a carrier on the wound surface, is requested.

Response: Thank you for your comment. We are sorry for this oversight. This has been corrected to 50 mg/kg. In paragraph 180 in the updated version, we added “which our laboratory used in previous research [1] …The hydrogel (20% Poloxamer407+6% Poloxamer188+74% H2O) was designed to become more gelatinous at temperatures above 37 ËšC, facilitating its retention on the wound”.

[1]Yu, F., D. Geng, Z. Kuang, S. Huang, Y. Cheng, Y. Chen, F. Leng, Y. Bei, Y. Zhao, Q. Tang, et al. "Sequentially releasing self-healing hydrogel fabricated with tgfβ3-microspheres and bfgf to facilitate rat alveolar bone defect repair." Asian J Pharm Sci 17 (2022): 425-34. 10.1016/j.ajps.2022.03.003.

3.In the "3.1 Spontaneous Differentiation of hPDLSCs into Myofibroblasts" section, clarification is sought regarding the rationale for measuring vimentin expression. Is vimentin upregulated or downregulated during myofibroblast differentiation? In Figure 1C, which represents western blotting, the Y-axis in the quantitative bar graph should be corrected to indicate relative protein expression for all proteins, not just COL1A1.

Response: We are grateful for this suggestion and apologize for this oversight. The Y-axis in the quantitative bar graph has been corrected.

4.In Figure 2D, the western blotting results show a subtle decrease in α-SMA with AGEs D8 treatment, but the quantitative bar graph does not fully support this observation. Additional western blotting images with corresponding data are requested for validation. Additionally, an explanation is needed for the absence of vimentin analysis in the cellular experiments involving AGEs in this section.

Response: Thank you for your comment. We carried out the WB of vimentin, and we have attached the data below.

5.If the statement "At D21, the wounds in the three groups were almost invisible and covered with hair, suggesting that all groups were close to complete healing" is accurate, then the bar graph in Figure 3C for D21 should approach 100%. Moreover, the healing percentages for D14 and D21 in Figure 3C are similar across all groups, around 80%, regardless of the treatment.

Response: We are grateful for this suggestion. We reviewed the data and made corrections.

6.Regarding Figure 5, an explanation is sought for why α-SMA is highest in the hPDLSCs group at D14 but lowest at D21.

Response: We are grateful for this suggestion. We noted in the discussion that MFs do not persist and highly express α-SMA during the whole process of wound healing. MF differentiation is delayed in diabetic wounds, and a high expression of α-SMA can still be observed at later time points as the healing time progresses. In normal wounds, MFs appear early in healing and decrease during tissue remodeling, but MF activity should be reduced or eliminated by apoptosis during late wound healing [1].

References:

  • Smith P C. Role of myofibroblasts in normal and pathological periodontal wound healing[J]. Oral Dis, 2018, 24(1-2): 26-29.

7.Figure 6 lacks legends; kindly provide the necessary legends for clarity.

Response: We are grateful for this suggestion and apologize for this oversight. The legend had been added.

Reviewer 3 Report

Comments and Suggestions for Authors

In this work, the authors study the differentiation of stem cells into myofibroblasts in diabetic wounds and the ability of human periodontal ligament stem cells to resist damage from AGEs in vitro and in vivo. The authors demonstrated the ability of hPDLSCs, ignoring damage from AGEs, to repair wounds. The authors hypothesize the use of stem cell therapy for wound healing and repair in clinical diabetic patients.

The following are some comments for the authors:     

The authors should add the molecular weight of the proteins for all western blotting images.

It would be helpful to view the original western blotting images. It may be better to present full membranes of western blots in unpublished materials as a power point presentation.

Figure 1 - Western blotting images of bands do not correspond to the calculation - all bands in the figure for a-MEM.

Figure 1 C - wb numbers are presented but mRNA is plotted, why?

Where is the difference between the mRNA in the plot of Fig. 1C and 1D? How can the authors explain such identical wb and qPCR results?

A legend for the color of the histogram of Figure 1D is required.

Figure 1D - is there statistical significance for COL1 between the blue and orange histogram for a-MEM?

Figure 2E - is there statistical significance between CTRL and AGE D4?

Figure 3B - authors should label the columns.

Figure 5 - need to indicate statistical significance in the graphs. Increase the font on the graphs.

Figure 6 - need a caption

Figure 6 - need a scale bar.

Figure 6 - authors should explain why the dapi intensity is different in all images?

Lines 391, 402, 437 - what does "Error!Reference source not found" means?

The authors should add a quantification for Figure 6.

Lines 329, 496 - control the font of the text.

The authors should explain why they did not also show the results for iNOS and CD206 with D7 (Figure 6) if the main significant effect for wound healing (Figure 3) they observed on D7?

Author Response

Reviewer #3:

1.In this work, the authors study the differentiation of stem cells into myofibroblasts in diabetic wounds and the ability of human periodontal ligament stem cells to resist damage from AGEs in vitro and in vivo. The authors demonstrated the ability of hPDLSCs, ignoring damage from AGEs, to repair wounds. The authors hypothesize the use of stem cell therapy for wound healing and repair in clinical diabetic patients.

The following are some comments for the authors:     

The authors should add the molecular weight of the proteins for all western blotting images.

Response: Thank you for your suggestion. We have added the molecular weight of the protein for the WB images in both Figure 1 and Figure 2.

It would be helpful to view the original western blotting images. It may be better to present full membranes of western blots in unpublished materials as a power point presentation.

Response: Thank you for your comment. We cut all WB membranes before the developing step due to the limitations of older experimental instruments. We can guarantee the authenticity of our raw data.

2.Figure 1 - Western blotting images of bands do not correspond to the calculation - all bands in the figure for a-MEM.

Response: Thank you for your comment and we apologize for this oversight. The corresponding error has been re-calculated and corrected.

3.Figure 1 C - wb numbers are presented but mRNA is plotted, why?

Response: Thank you for your comment. We are sorry for this oversight. We have corrected the Y-axis in Figure 1C.

4.Where is the difference between the mRNA in the plot of Fig. 1C and 1D? How can the authors explain such identical wb and qPCR results?

Response: Thank you for your comment. We apologize for mixing up the qPCR results with the WB results, leading to repetition in this figure. We have corrected the WB results as followed:

5.A legend for the color of the histogram of Figure 1D is required.

Response: We are grateful for the suggestion and apologize for this oversight. The legend for Figure 1D has been added.

6.Figure 1D - is there statistical significance for COL1 between the blue and orange histogram for a-MEM?

Response: Thank you for your comment. They were statistically significant, and we have corrected this in Figure 1D.

7.Figure 2E - is there statistical significance between CTRL and AGE D4?

Response: Thank you for your comment. We have re-annotated the statistical significance as shown below:

8.Figure 3B - authors should label the columns.

Response: We are grateful for this suggestion and apologize for this oversight. Columns have been added.

9.Figure 5 - need to indicate statistical significance in the graphs. Increase the font on the graphs.

Response: We are grateful for this suggestion. The statistical significance has been indicated in the chart and the font on the graph has been added.

10.Figure 6 - need a caption

Response: We are grateful for this suggestion and apologize for this oversight. The caption had been added at L278:“Immunofluorescence staining of iNOS, CD206, and F4/80 in wound tissue on D14 and D21.”

11.Figure 6 - need a scale bar.

Response: We are grateful for this suggestion. Scale bars were included in every picture, but they may have been too small, so we have added a larger scale bar to Figure 6.

12.Figure 6 - authors should explain why the dapi intensity is different in all images?

Response: Thank you for your comment. We used the same parameter to scan those sections and the DAPI intensity seems different because some pictures have been compressed into a smaller format. We have re-uploaded the high resolution images to correct this mistake.

13.Lines 391, 402, 437 - what does "Error!Reference source not found" means?

Response: We are grateful for the suggestion. The cited literature has made some errors in different versions, and the corresponding positions have now been corrected.

14.The authors should add a quantification for Figure 6.

Response: We are grateful for the suggestion. We are sorry for this oversight. Quantification has been added.

15.Lines 329, 496 - control the font of the text.

Response: We are grateful for the suggestion. We are sorry for this oversight. The text font at that location has been corrected.

16.The authors should explain why they did not also show the results for iNOS and CD206 with D7 (Figure 6) if the main significant effect for wound healing (Figure 3) they observed on D7?

Response: Thank you for your critical comment. Our research of this part is intended to observe the macrophage polarization in the late stage of wound healing influenced by hPDLSCs, not only the inflammatory condition that has been shown in HE staining results. Moreover, some researches have claimed that M2 can’t increase during Day0-Day10 in diabetic wounds[1-2], so we think the difference between D14 and D21 can illustrate what we want to present to readers.

[1]Louiselle A E, Niemiec S M, Zgheib C, et al. Macrophage polarization and diabetic wound healing[J]. Transl Res, 2021, 236: 109-116.

[2]Li M, Hou Q, Zhong L, et al. Macrophage related chronic inflammation in non-healing wounds[J]. Front Immunol, 2021, 12: 681710.

Round 2

Reviewer 1 Report

Comments and Suggestions for Authors

1.      “The title needs to be revised to better reflect the data of MS”.

My message was that the Title need to be revised, not grammatically corrected.

 In fact, the MS is consisted of two parts. First one is dealing with in vitro characterization of cell product – PDLSCs. The second one is describing the results of application of this cell product in diabetic wounds in mice. The  challenge is that the Authors did not follow the fate of the injected cells. They just have characterized  the host endogenous cell populations no making differences between the injected exogenous PDLSCs and host cells in the wound.

It had to be reflected in the Title.

2.      “It is generally accepted that exogenous multipotent mesenchymal stem/stromal cells (MSCs), to which PDLSCs belong, are retained in the body for a short time before being eliminated. The potential of MSCs for repair through direct differentiation into myofibroblasts   is a matter of speculation. Therefore, it is important to distinguish between in vitro studies and in vivo differentiation observations when discussing the contribution of MSCs or PDLSCs in wound healing”

Response: Thank you for your comment. hPDLSCs with myofibroblastic differentiation were cultured in H-DMEM medium in vitro, and the 9th passage of differentiated cells were injected into the wound area of STZ rats to observe the effect on wound healing.

The Authors have ignored my main message concerning the fate of injected cells. 

Please, explain.

Author Response

  1. “The title needs to be revised to better reflect the data of MS”.

My message was that the Title need to be revised, not grammatically corrected. In fact, the MS is consisted of two parts. First one is dealing with in vitro characterization of cell product – PDLSCs. The second one is describing the results of application of this cell product in diabetic wounds in mice. The  challenge is that the Authors did not follow the fate of the injected cells. They just have characterized  the host endogenous cell populations no making differences between the injected exogenous PDLSCs and host cells in the wound.

It had to be reflected in the Title.

 Response: Thank you for your comment and we apologize for this oversight. We revised the title into “Human periodontal ligament stem cells(hPDLSCs) differentiated into myofibroblasts and its application in accelerated diabetic wounds healing”. Does it conform with the MS?

  1. “It is generally accepted that exogenous multipotent mesenchymal stem/stromal cells (MSCs), to which PDLSCs belong, are retained in the body for a short time before being eliminated. The potential of MSCs for repair through direct differentiation into myofibroblasts   is a matter of speculation. Therefore, it is important to distinguish between in vitro studies and in vivo differentiation observations when discussing the contribution of MSCs or PDLSCs in wound healing”

Response: Thank you for your comment. hPDLSCs with myofibroblastic differentiation were cultured in H-DMEM medium in vitro, and the 9th passage of differentiated cells were injected into the wound area of STZ rats to observe the effect on wound healing. 

The Authors have ignored my main message concerning the fate of injected cells. 

Please, explain.

Response: Thank you for your suggestion. We have simulated a diabetic environment in vitro induced by AGEs and we observed the hPDLSCs can spontaneously differentiate into myofibroblasts in this environment, then we use the culture method that hPDLSCs can spontaneously differentiate into myofibroblasts to culture hPDLSCs to the proper passage and injected them into the diabetic wound area, this process can be understood as the induced culture of hPDLSCs to treat the diabetic wounds. Although we didn’t trace the injected hPDLSCs, we did observe that the hPDLSCs treated group have more myofibroblasts than other groups that can indirectly illustrate hPDLSCs can spontaneously differentiate into myofibroblasts also in vivo. We are willing to add experiments to trace the fate of hPDLSCs if necessary.

Reviewer 3 Report

Comments and Suggestions for Authors

The authors addressed almost all comments and revised their manuscript accordingly.

Please, add code number of used antibody in the section Methods and correct in all the text - Error! Reference source not found

Author Response

  1. The authors addressed almost all comments and revised their manuscript accordingly.

Please, add code number of used antibody in the section Methods and correct in all the text - “Error! Reference source not found”.

Response: Thank you for your comment and we apologize for this oversight. We have made corrections to the formatting errors in the literature citation and added code number of used antibody in the section Methods.